# A Numerical Simulation Method Considering Solid Phase Transformation and the Experimental Verification of Ti6Al4V Titanium Alloy Sheet Welding Processes

**DOI:** 10.3390/ma15082882

**Published:** 2022-04-14

**Authors:** Yu Li, Jia-Yi Hou, Wen-Jian Zheng, Zheng-Quan Wan, Wen-Yong Tang

**Affiliations:** 1China Ship Scientific Research Center, Wuxi 214082, China; 21634074@zju.edu.cn (J.-Y.H.); zqwans@eyou.com (Z.-Q.W.); 2Department of Chemical Machinery, College of Mechanical Engineering, Zhejiang University of Technology, Hangzhou 310000, China; zwj0322@zjut.edu.cn; 3School of Naval Architecture, Ocean & Civil Engineering, Shanghai Jiao Tong University, Shanghai 200011, China; wytang@sjtu.edu.cn

**Keywords:** titanium alloy, laser beam welding, tungsten inert gas welding, solid-state phase transformation, residual stresses, numerical simulation

## Abstract

A prediction model of the welding process of Ti-6Al-4V titanium alloy was established by using the finite element method, which was used to evaluate the phase composition, residual stress and deformation of the welded joints of Ti-6Al-4V sheets with different processes (including tungsten inert gas welding, TIG, and laser beam welding, LBW). The Ti-6Al-4V structures of TIG welding and LBW are widely used in marine engineering. In order to quantitatively study the effects of different welding processes (including TIG welding and LBW) on the microstructure evolution, macro residual stress and deformation of Ti6Al4V titanium alloy sheets during welding, a unified prediction model considering solid-state phase transformation was established based on the ABAQUS subroutine. In this paper, LBW and TIG welding experiments of 1.6 mm thick Ti-6Al-4V titanium alloy sheets were designed. The microstructure distribution of the welded joints observed in the experiment was consistent with the phase composition predicted by the model, and the hardness measurement experiment could also verify the phase composition and proportion. From the residual stress measured by experiment and the residual stress and deformation calculated by finite element simulation of LBW and TIG weldments, it is concluded that the effect of phase transformation on residual stress is mainly in the weld area, which has an effect on the distribution of tensile and compressive stress in the weld area. The overall deformation of the welded joint is mainly related to the welding process, and the phase transformation only affects the local volume change of the weld seam. Importantly, the phase composition and residual stress, which are scalar fields, calculated by the established model can be introduced into the numerical analysis of structural fracture failure as input influence factors.

## 1. Introduction

The Ti-6Al-4V titanium alloy is widely used in marine engineering because of its excellent strength, ductility and corrosion resistance. Tungsten inert gas welding (TIG) and laser beam welding (LBW) are the two most widely used welding processes in the manufacturing of offshore structures. During the implementation of projects, the welding processes is selected according to the requirements of the environment and service conditions, comprehensively considering the cost, efficiency and the quality of welded joints. Therefore, it is crucial to establish an evaluation system for TIG welding and LBW processes for the Ti-6Al-4V titanium alloy.

In fact, there have been many research findings on TIG welding and LBW of titanium alloy. Short et al. (2009) [1] found that the power density of TIG welding is much lower than that of LBW welding, and TIG weldments have wider heat-affected zones (HAZs) and higher residual stress and deformation levels. Gao et al. (2013) [2] found that for the welding of Ti-6Al-4V titanium alloy plates, TIG welding leads to high residual deformation and coarse grain structure levels, while LBW will have lower residual deformation and higher Vickers hardness values. Zhang et al. (2005) [3] showed that the residual stress distribution of LBW of titanium alloy sheets is similar to that of TIG welding; compared with TIG welding, the residual stress distribution of LBW welding is narrower, and the maximum residual stress can reach 80% of the yield strength. Goldak and Akhlaghi (2005) [4] posit that local heating of welding will lead to uneven thermal expansion. The large difference in thermal expansion coefficients of the titanium alloy α phase and β phase makes the local non-uniform expansion more obvious, resulting in excessive residual stress, which leads to the buckling deformation of the weldment sheet.

The welding processes of titanium alloy affect the microstructure of welded joints, and the microstructure directly affects its mechanical properties. Therefore, it is very important to understand and control the evolution of microstructures. In welding, the Ti-6Al-4V titanium alloy undergoes continuous heating and cooling with mutual transformation between the α and β phases. According to the research of Elmer (2005) [5] on the Ti-6Al-4V titanium alloy, when the heating temperature exceeds the β phase transition point (980 °C), the α phase completely transforms into the β phase. In cooling from the temperature above the β phase transition point (980 °C), when the cooling rate is higher than 410 °C/s, the β phase is completely transformed into a martensite α′ phase; when the cooling rate is from 20 °C/s to 410 °C/s, a partial transformation of a similar structure can be observed; and when the cooling rate is less than 20 °C/s, the β phase completely transforms into the α phase.

The microstructure of the Ti-6Al-4V titanium alloy in welding is extremely complex. In order to quantitatively study the relationship between the microstructure and macro mechanics, it is necessary to simplify the complex microstructure evolution of Ti-6Al-4V into a general solid-state phase transformation mathematical model. In recent years, some models have been developed for solid phase transformations, but these models are mainly applicable to steel. The JMAK equation (Johnson Mehl, Avrami and Kolmogorov) has been successfully used to describe the diffusion phase transformation during welding, especially for the transformation process of the β→α phase of titanium alloys. Although the formation mechanism of martensite α′ in titanium alloy welding is not clear, the K–M (Koistinen Marburger) equation has been used to describe this non-diffusion phase transformation. Elmer et al. (2005) [5] studied the thermodynamic properties of the α phase and β phase of titanium alloys. They experimentally measured the thermal expansion coefficient of each phase with temperature, and the unit lattice volume of each phase varying with temperature. It was found that the thermal expansion coefficient and the unit lattice volume of the α phase and β phase vary greatly with temperature. Thus, the phase composition and phase proportion of Ti-6Al-4V welded joints can be predicted according to the solid-state phase transformation model, and the thermodynamic parameters of different phases have been measured experimentally, so the model can be established to quantitatively describe the relationship between microstructure and residual stress.

The research on the welding of Ti-6Al-4V titanium alloy is very extensive, including the research on the microstructure and phase transformation model of welding, the research on welding residual stress and deformation and the research on the fracture toughness of welded joints. The correlation and causality between these studies can also be found. Thus, answering the question of how to unify these achievements into an engineering evaluation system is a meaningful and innovative task. In the current commercial software, SYSWELD is very representative in welding and heat treatment. KIK Tomasz et al. [6,7,8] used SYSWELD software to predict the residual stress and deformation of MIG welding of aluminum alloy sheets and simulated the multi-pass welding process of duplex steel. They used the metallurgical calculation module to calculate the phase distribution in the weld seam and the recrystallization phase distribution in the heat-affected zone. ABAQUS software can also use subroutines to calculate the evolution of phase composition during welding. Based on the ABAQUS subroutine, this paper innovatively uses the finite element method to unify the welding process, microstructure, residual stress and deformation into one model and achieve the calculation and prediction of the phase and the residual stress and deformation with scalar field. It can not only determine the welding process by predicting the microstructure and residual stress of the welded joint, but also can introduce these scalar fields into the numerical analysis of structural fracture failure as input influence factors.

Based on the above research findings of titanium alloy welding, we attempted to establish a numerical method in this paper to quantitatively study the effects of different welding processes (TIG welding and LBW) on the phase composition (mainly α phase, martensite α′ phase and β phase), phase volume fraction and residual stress and deformation of Ti-6Al-4V titanium alloy welded joints. In order to verify and compare, TIG and LBW experiments of 1.6 mm thick Ti-6Al-4V titanium alloy sheets were designed, and the experimental results were compared with the simulation results of the numerical calculation method. The procedure is as follows: the correctness of Ti-6Al-4V phase transformation model and its applicability to welding process were verified by comparing the microstructure observations of welded joints with the calculated phase volume fraction and the hardness measurement value with the calculated phase volume fraction distribution; by comparing the measured residual stress of LBW and TIG-welded joints with the simulation results, the rationality of the calculations of residual stress and deformation considering solid-state phase transformation was verified. In extension, because the phase volume fraction and residual stress field variables can express characteristics of the welding process, this research can introduce welding process factors into the numerical analysis of structural fracture failure, which can provide evaluation criteria for the selection of welding processes in engineering applications.

## 2. Materials and Methods

### 2.1. Material Performance of Ti6Al4V

The Ti-6Al-4V titanium alloy is one of the most commonly used titanium alloys. It is widely used in marine engineering because of its excellent strength, toughness and corrosion resistance. The alloy consists of two phases, the hexagonal close-packed (HCP) α phase and the body-centered cubic (BCC) β phase. At room temperature, the alloy consists of 80~95% α phase; the rest is β phase [9]. The material used in this study was rolled, annealed titanium alloy Ti-6Al-4V plate with a thickness of 1.6 mm. The chemical composition of Ti-6Al-4V is shown in Table 1.

The thermo-physical material properties of Ti-6Al-4V varying with temperature used for calculation came from the literature, and the material properties are shown in Figure 1. The elastic modulus, Poisson’s ratio and density values were obtained by JMatPro software, and the specific heat and thermal conductivity values were obtained by Boivineau et al. (2006) [10]. According to the values given by Elmer et al. (2005) [5], the solid temperature (T_solid_) of Ti-6Al-4V is 1878 K, the β phase transition temperature (T_β_) is equal to 1248 K, the liquidus temperature (T_liquidus_) is equal to 1928 K and the boiling point (T_b_) is equal to 3315 K. As can be seen in Figure 1, the properties’ sudden changes occur at the temperatures T_β_ (β-phase transition temperature) and T_liquidus_ (liquidus temperature); this is due to the latent heat effect of energy release or absorption during solidification or melting caused by phase transformation. The latent heat of melting (Δh_fus_) is equal to 286 kJ/kg and the latent heat of evaporation (Δh_evap_) is equal to 9830 kJ/kg.

In order to obtain the stress–strain relationship of the Ti-6Al-4V titanium alloy at different temperatures, a tensile test was carried out with a strain rate of 0.001 s^−1^ in a wide temperature range (20~1000 °C), and the tensile properties of Ti-6Al-4V varying with temperature were obtained, as shown in Figure 2 [9].

### 2.2. The LBW and TIG Welding Processes of Ti6Al4V Titanium Alloy

LBW is characterized by fast welding speeds, small heat-input-per-unit times, small range of weld-heat-affected zones, narrow weld seams, fast cooling speeds, small change of weld-seam properties, high hardness of weld seams, etc. TIG welding is characterized by slow welding speeds, large heat-input-per-unit times, large fusion zones and heat-affected areas and easy welding deformation. The characteristics of the two welding processes determine their differences of microstructure and mechanical properties in the weld seam. 

In this paper, the butt welding of a Ti-6Al-4V titanium alloy sheet with a length of 100 mm, width of 80 mm and thickness of 1.6 mm are taken as the research object, welded by LBW and TIG welding respectively. The welding process and equipment are shown in Figure 3. The laser used for laser welding was IPG/YLS-4000S2T, which is a pulse laser, and the wavelength of the laser was 1065 nm, the M^2^ of the multimode beam quality factor of the laser was 2.05, the laser focal length was 340 mm, the beam spot diameter was 1 mm, the position of focal plane with respect to surface of the sample was 0 mm and the gas used for protection was argon. The welding parameters of the TIG welding and LBW of the Ti6Al4V titanium alloy are shown in Table 2 and Table 3. 

Figure 4 shows the surface appearance of TIG and LBW weldments after welding. Due to the use of inert gas for front and rear purging, the weldment surface is smooth, free of oxides, cracks and pores. The TIG weld seam is wider than the LBW weld seam, and the range of the heat-affected area is significantly wider, mainly because TIG welding has a higher heat-input-per-unit time.

### 2.3. Experiment on Microstructure and Macro Mechanical Properties

#### 2.3.1. Hardness Measurement and Microstructure Observation 

The Vickers hardness was measured by using the micro-hardness tester VMH-I04 (Walter Uhl techn. Mikroskopie GmbH & Co. KG, Hesse, Germany) from the middle of the weld seam outward in the middle of the specimen plate, perpendicular to the direction of the welding direction, and the applied pressure was HV 0.2 (1.96 N) with a dwell time of 10 s.

Metallographic samples were obtained by wire cutting in the middle of the weldment. Firstly, the metallographic sample was mechanically ground with water abrasive paper and then polished by a metallographic grinder with 1 μm Struers diamond suspension paste. After polishing, it was etched with Kroll reagent (92 mL distilled water, 6 mL HNO_3_ and 2 mL hydrofluoric acid) and cleaned with 0.2% hydrofluoric acid to show the microstructure, and the microstructure was observed under the metallographic microscope OLYMPUS GX71 (Olympus Corporation, Shenzhen, China).

#### 2.3.2. Measurement of Residual Stresses

At present, the most widely used residual stress measurement method in weldments and castings is the hole drill strain measurement (HDSM). The experimental device for measuring residual stress by the HDSM is shown in Figure 5. It is a simple and reliable method. Because its drilling diameter and depth are small, it produces limited damage to the workpiece. It is suitable for residual stress measurement of various alloys [11] (Rossini et al., 2012). Based on the theory of elasticity, the principle of measuring residual stress by this method is that through drilling a hole at the measuring point the stress is released and the strain is redistributed; the release strain can be measured from the strain gauge fixed around the hole, and the residual stress can be obtained according to the stress–strain relationship. This method has high accuracy and is widely used.

The arrangement of strain-measuring points on the weldment is shown in Figure 5b. The boundary conditions at both ends of the welding centerline are the same, and the welding is a symmetrical problem. Therefore, the distribution of residual stress along the welding direction can be considered as symmetrical. When arranging the measuring points of welding residual stress, only one measuring point needs to be arranged at the symmetrical position, and the number of measuring points can be reduced by half.

## 3. Numerical Simulation Method of Welding Processes

The prediction of welding residual stress and deformation is still a complex and difficult task because it involves the establishment of a complex, nonlinear elastoplastic analysis model. It must consider the moving heat source, material properties varying with temperature, plastic flow, volume expansion during metallurgical transformation, filler metal deposition and thermal and mechanical boundary conditions during welding. This means that welding simulation requires a lot of computing resources and a lot of data storage. Therefore, it is necessary to make some assumptions, approximations and simplifications to ensure that the prediction is accurate enough.

### 3.1. Numerical Calculation of Welding Temperature Field

Selection of heat source model for different welding technologies

The double-ellipsoidal heat source [12] acting on the workpiece is distributed along the welding direction, as shown in Figure 6a. The model is divided into front and rear parts. The heat flow distribution along the front and rear parts is shown in Equations (1) and (2):(1)$$q_{f}\left(x,y,z\right)=\frac{6\sqrt{3}f_{f}Q}{a_{1}bc\pi \sqrt{\pi }}\exp\left(-\frac{3x^{2}}{a_{1}{}^{2}}-\frac{3y^{2}}{b^{2}}-\frac{3z^{2}}{c^{2}}\right),x\geq 0$$
(2)$$q_{r}\left(x,y,z\right)=\frac{6\sqrt{3}f_{r}Q}{a_{2}bc\pi \sqrt{\pi }}\exp\left(-\frac{3x^{2}}{a_{2}{}^{2}}-\frac{3y^{2}}{b^{2}}-\frac{3z^{2}}{c^{2}}\right),x<0$$
where $q_{f}\left(x,y,z\right)$ and $q_{r}\left(x,y,z\right)$ are the heat flow distribution in the volume of the front and rear half ellipsoids, W/m^3^; $a_{1}$ is the front half axis length of the molten pool; $a_{2}$ is the rear half axis length of the molten pool; b is the width of the molten pool and c is the depth of the molten pool, m. Additionally, $f_{f}$ and $f_{r}$ are the heat flow density distribution coefficients of the front and rear hemispheres of the double ellipsoids; the sum of the two remains unchanged, $f_{f}+f_{r}=2$.

The double-ellipsoidal heat source is widely used in the numerical simulation of TIG welding. It takes the molten pool as the expression object, but it is difficult to determine the shape of the high-temperature molten pool. The selection of heat source model parameters needs repeated trial calculations to obtain the appropriate shape of molten pool.

The heat source of LBW is a three-dimensional Gaussian conical heat source model [9], as shown in Figure 6b. Its radial power density distribution is Gaussian distribution, and its axial distribution is linear distribution. The heat flow on the top surface of the conical heat source is the largest, and the heat flow on the bottom surface is the smallest:(3)$$q_{r}\left(r,z\right)=q_{0}e^{-3{\left(\frac{r}{r_{0}}\right)}^{2}},\;\;\;0\leq r\leq r_{0}\left(z\right),-H\leq z\leq 0$$
where $r_{0}$ is the radius of the heat source at a specific height, *Z*; $q_{0}$ is the maximum volume power density and r is the radius of the current internal point, according to Equation (4):(4)$$r={\left[{\left(x-x_{0}\right)}^{2}+{\left(y-y_{0}-vt\right)}^{2}\right]}^{\frac{1}{2}}$$

The distribution parameter $r_{0}$ decreases linearly from the top to the bottom of the conical region, as shown in Equation (5).
(5)$$r_{0}\left(z\right)=r_{e}-\left(r_{e}-r_{i}\right)\frac{\left(z_{e}-z\right)}{z_{e}-z_{i}}$$
where $z_{e}$ is the top surface and $z_{i}$ is the bottom surface of the cone’s area. By integrating the volumetric heat flow and taking it as the total heat input, the maximum volumetric power density can be obtained, as shown in Formula (6).
(6)$$Q=\underset{U}{∭}q_{r}\left(x,y,z\right)dxdydz=\underset{U}{∭}q_{r}\left(r,\theta \right)rdrd\theta dh=\int_{0}^{H}\int_{0}^{2\pi }\int_{0}^{r_{0}}q_{0}e^{-\frac{3r^{2}}{r_{0}^{2}}}rdrd\theta dh$$
$$q_{0}=\frac{9\eta Qe^{3}}{\pi \left(e^{3}-1\right)\left(z_{e}-z_{i}\right)\left(r_{e}^{2}+r_{e}r_{i}+r_{i}^{2}\right)}$$
where $H=z_{e}-z_{i}$, $h=z-z_{i}$, *e* is the base of the natural logarithm and η is the heat source efficiency. The heat source efficiency or energy transfer efficiency is defined as the ratio of the energy absorbed by the irradiated material to the laser output power.

Welding heat transfer process.

In the welding process, the governing equation of instantaneous heat conduction is shown in Equation (7) [13].
(7)$$\rho \left(T\right)c\left(T\right)\frac{\partial T}{\partial t}=\frac{\partial }{\partial x}\left[k\left(T\right)\frac{\partial T}{\partial x}\right]+\frac{\partial }{\partial y}\left[k\left(T\right)\frac{\partial T}{\partial y}\right]+\frac{\partial }{\partial z}\left[k\left(T\right)\frac{\partial T}{\partial z}\right]+Q_{h}$$
where ρ is density, kg/m^3^; c is specific heat capacity, J/(kg·K); T is temperature, K; k is thermal conductivity, W/(m·K) and $Q_{h}$ is the heat generation rate, W/m^3^.

The governing equations of heat radiation and heat convection on the workpiece surface during welding is expressed as follows:(8)$$q_{c}=h_{f}\left(T-T_{0}\right)$$
(9)$$q_{r}=\varepsilon \sigma \left(T^{4}-T_{0}^{4}\right)$$
where $h_{f}$ is the air heat transfer coefficient, W/m^2^/°C; $T_{0}$ is the ambient temperature; σ is the Stefan–Boltzmann coefficient and ε is the radiation coefficient.

### 3.2. Numerical Calculation of Welding Residual Stress

In computational welding mechanics, the total strain can be divided into several parts. In order to facilitate the implementation of the finite element method, it can be written in incremental form as follows [9]:(10)$$\Delta \varepsilon =\Delta \varepsilon^{e}+\Delta \varepsilon^{p}+\Delta \varepsilon^{T}+\Delta \varepsilon^{\Delta \mathrm{Vol}}+\Delta \varepsilon^{\mathrm{Trp}}$$
where Δε is the total strain increment; $\Delta \varepsilon^{e}$ and $\Delta \varepsilon^{p}$ are the elastic strain increment and plastic strain increment, respectively; $\Delta \varepsilon^{T}$ is the thermal strain increment; $\Delta \varepsilon^{\Delta \mathrm{Vol}}$ is the volume change increment and $\Delta \varepsilon^{\mathrm{Trp}}$ is the phase-transformation-induced plastic strain increment. The volume strain increment $\Delta \varepsilon^{\Delta \mathrm{Vol}}$ is mainly a result of the different crystal structures of the β and α phases; the volume change is caused by mutual transformation in the process of heating or cooling. Generally speaking, the magnitude of the plastic strain increment $\Delta \varepsilon^{\mathrm{Trp}}$ induced by phase transformation is smaller than that of other terms.

In the previous analysis, the thermal strain is calculated according to the average thermal expansion coefficient, which is defined varying with temperature, but the strain calculated by this method ignores the volume shrinkage or expansion effect caused by the change of material microstructure. At this time, the thermal strain is not only dependent on the temperature change, but also related to the temperature change history of the material during heating and cooling. As shown in Figure 7a, the linear coefficients of thermal expansion of the α phase and β phase with temperature were obtained from JMatPro. After obtaining the volume fraction of each phase at arbitrary temperatures, the average thermal expansion coefficient of the material can be calculated by the linear superposition principle [14]:(11)$$\lambda =f_{\alpha }\lambda_{\alpha }+f_{\beta }\lambda_{\beta }$$
where $f_{\alpha }$ is the α phase volume fraction, $f_{\beta }$ is the β phase volume fraction, $\lambda_{\alpha }$ is the average thermal expansion coefficient of the α phase and $\lambda_{\beta }$ is the average thermal expansion coefficient of the β phase.

During the finite element calculation, the thermal strain is carried out in incremental form:(12)$$\varepsilon_{i}^{T}=\lambda_{f}\left(T_{f}-T_{0}\right)-\lambda_{I}\left(T_{I}-T_{0}\right)$$
(13)$$\varepsilon_{i-1}^{T}=\lambda_{s}\left(T_{s}-T_{0}\right)-\lambda_{I}\left(T_{I}-T_{0}\right)$$
(14)$$\Delta \varepsilon^{T}=\varepsilon_{i}^{T}-\varepsilon_{i-1}^{T}$$
where $T_{s}$ is the temperature at the start time, $T_{f}$ is the temperature at the end time, $T_{0}$ is the reference temperature and the initial temperature $T_{I}$ is the temperature corresponding to the thermal strain of 0, which is 20 °C. The terms $\varepsilon_{i}^{T}$ and $\varepsilon_{i-1}^{T}$ are the thermal strain values corresponding to incremental steps i and i−1, respectively. From the above formula, the strain increment $\Delta \varepsilon^{T}$ has nothing to do with the initial temperature $T_{I}$ but is related to the reference temperature $T_{0}$.

During the heating of Ti-6Al-4V, because there are some β phase elements in the microstructure of the PM, the volume of the β phase is increased through the diffusion and migration of the α/β interface to achieve the transformation from α + β to β. The volume ratio of α phase and β phase crystal units is similar. Therefore, the volume effect of the Ti-6Al-4V phase transition is modeled by considering the change of volume ratio of different phases, which is determined by the lattice parameters of different phases. Elmer et al. (2005) [8] measured lattice parameters of the α phase and β phase in real time by synchronous X-ray diffraction. Thus, the unit lattice volume of each phase and the phase-transformation-induced volumetric strain are determined, as shown in Figure 7b. Therefore, by multiplying the volume fraction of each microstructure phase and its respective unit crystal volume as a function of the welding temperature, the volume strain in the phase transformation caused by the different unit crystal volumes of the two phases is calculated.

### 3.3. Solid Phase Transformation of Titanium Alloy during Welding

Theoretical model of solid phase transformation

The phase transformation of titanium alloy welding can be regarded as two homogeneous transformations: one is the transformation of the heating process and the other is the transformation of the cooling process. The phase transitions of α→β and β→α are considered as two mutually inverse diffusion phase transitions, which is described by the Johnson–Mehl–Avrami equation [15]:(15)$$f\left(t,T\right)=\left[1-\exp\left(-k\left(T\right)t^{n}\right)\right]f^{\mathrm{eq}}\left(T\right)$$
where f is the volume fraction of the product phase at time t, $f^{\mathrm{eq}}$ is the equilibrium volume fraction of the product phase at temperature T and temperature dependent k(T) and temperature independent n are the parameters of the JMAK equation defining the phase transition kinetics.

In order to extend the isothermal transformation process of the above equation to the non-isothermal transformation process of welding, the Scheil additivity rule is used and the virtual time is introduced. The virtual time is shown in Equation (16):(16)$$t_{0}^{*}={\left[\frac{1}{k\left(T_{1}\right)}\ln\frac{f^{\mathrm{eq}}\left(T_{1}\right)}{f^{\mathrm{eq}}\left(T_{1}\right)-f\left(t_{0},T_{0}\right)}\right]}^{\frac{1}{n}}$$

The modified JMAK equation is shown in Equation (17):(17)$$f\left(t_{1},T_{1}\right)=\left[1-\exp\left(-k\left(T_{1}\right){\left(t_{0}^{*}+t_{1}-t_{0}\right)}^{n}\right)\right]f^{\mathrm{eq}}\left(T_{1}\right)$$
where $t_{0}^{*}$ is the virtual time required to achieve the isothermal transformation of $f\left(t_{0},T_{0}\right)$ at temperature $T_{1}$ and $f^{\mathrm{eq}}\left(T_{1}\right)$ is the phase equilibrium fraction of the product phase at temperature $T_{1}$.

During rapid cooling, the process of the β phase transformation to the martensite α′ phase is a non-diffusion transformation, which is described by the Koistinen–Marburger equation:(18)$$f_{M}\left(t,T\right)=1-\exp\left(-k_{M}\left(M_{S}-T\right)\right)$$
where $f_{M}$ is the volume fraction of the martensite α′ phase, $k_{M}$ is a material dependent parameter and $M_{S}$ is the martensite starting temperature.

2Solid phase transformation model of titanium alloy welding

In the heating stage, Ti-6Al-4V experiences rapid diffusion-controlled transformation from the α phase to the β phase. The diffusion-controlled transition is described by the JMAK equation. In order to avoid the morphological differences of globular, Widmanstatten structure, basket structure and grain boundary α phase types, a simplified single α phase is assumed. During heating, the β phase volume fraction is shown in Equation (19):(19)$$f_{\beta }\left(t_{1},T_{1}\right)=\left[1-\exp\left(-k_{\alpha \to \beta }\left(T_{1}\right){\left(t_{0}^{*}+t_{1}-t_{0}\right)}^{n_{\alpha \to \beta }}\right)\right]f_{\beta }^{\mathrm{eq}}\left(T_{1}\right)\left(f_{\beta }\left(t_{0},T_{0}\right)+f_{\alpha }\left(t_{0},T_{0}\right)\right)$$
where $t_{0}^{*}$ is the virtual time required to achieve the $f_{\beta }\left(t_{0},T_{0}\right)$ through the isothermal transformation at temperature $T_{1}$ and $f_{\beta }^{\mathrm{eq}}$ is the equilibrium fraction of the β phase.

In the cooling stage, the cooling rate determines the transformation kinetics and the formed phase. Below the temperature of the β phase transition point, when the cooling rate is less than 20 °C/s, the β phase decomposes into the α phase by nucleation and diffusion reactions:(20)$$f_{\alpha }\left(t_{1},T_{1}\right)=\left[1-\exp\left(-k_{\beta \to \alpha }\left(T_{1}\right){\left(t_{0}^{*}+t_{1}-t_{0}\right)}^{n_{\beta \to \alpha }}\right)\right]f_{\alpha }^{\mathrm{eq}}\left(T_{1}\right)\left(f_{\beta }\left(t_{0},T_{0}\right)+f_{\alpha }\left(t_{0},T_{0}\right)\right)$$
where $t_{0}^{*}$ is the virtual time required to achieve the $f_{\alpha }\left(t_{0},T_{0}\right)$ through isothermal transformation at temperature $T_{1}$ and $f_{\alpha }^{\mathrm{eq}}$ is the equilibrium fraction of α phase.

According to the research of Ahmed and Rack (1998) [16], if the rapid cooling rate is greater than 410 °C/s, the complete martensite α′ will be formed and the α phase produced through diffusion transition will be inhibited:(21)$$\begin{array}{ll} f_{\alpha^{\prime }}\left(t_{1},T_{1}\right) & =f_{\alpha^{\prime }}\left(t_{0},T_{0}\right)+\Delta f_{\alpha^{\prime }}\\ & =\left[1-\exp\left(-k_{\beta \to \alpha^{\prime }}\left(M_{S}-T\right)\right)\right]\left(f_{\beta }\left(t_{0},T_{0}\right)+f_{\alpha^{\prime }}\left(t_{0},T_{0}\right)\right) \end{array}$$
where $f_{\alpha^{\prime }}$ is the volume fraction of α′ phase and $f_{\beta }^{\mathrm{eq}}$ is the equilibrium volume fraction of β phase.

In the case of moderate cooling rates between 20 and 410 °C/s, it begins to cool from the temperature above the β phase transition point, which will lead to the massive α phase transition at the grain boundary of the β phase and partial transformation of martensite α′ in the crystal of the primary β grain boundary. When the temperature is higher than the martensitic transformation temperature *M*_s_, the β phase transformation to the α phase at the grain boundary first occurs; when the temperature decreases below the martensitic transformation temperature *M*_s_, martensite α′ is formed in the crystals of the β phase.

The JMAK equation parameters $k_{\beta \to \alpha }$ and $n_{\beta \to \alpha }=2.5$ used in this study are based on the Ti-6Al-4V TTT diagram of Kelly, which is obtained by modeling the JMatPro TTT curve fitting, as shown in Figure 8 [17].

### 3.4. Implementation of Numerical Simulation and Parameters Calibration

During welding, the welded joint is generally divided into three areas: the fusion zone (FZ), heat-affected zone (HAZ) and parent metal (PM). In the zone close to the weld seam and heat source, due to the greatly varying ranges of temperature gradients and stress gradients, the mesh of this zone needs to be refined; in the zone far from the weld seam and heat source, the temperature gradient and stress gradient change little. The mesh of this zone is roughly divided, which can greatly shorten the simulation time. In this paper, ABAQUS software is used for analysis. The welding finite element models of LBW and TIG welding are shown in Figure 9, respectively. The welding area is fine mesh, and the minimum element size is 0.5 mm × 0.5 mm, and then the grid transition form of 1:2 to connect the grids of three adjacent areas is used. After meshing, the number of elements of LBW is 95,200 and the number of elements of TIG welding is 95,200. Because the area of the heat-affected zone and fusion zone of TIG welding is larger than that of LBW, the area of refined mesh is larger. The grid division in the above way not only ensures the accuracy of the results of the welding area, but also improves the calculation efficiency.

In the calculation process, 8-node DC3D8 linear element is used in thermal analysis, and 8-node C3D8R linear element is used in mechanical analysis. The purpose of the thermal analysis process is to obtain an accurate temperature evolution history curve in the finite element calculation, which will be the initial input condition for the subsequent stress analysis model for the calculation of welding residual stress. In the stress analysis, since the chemical composition and mechanical properties of the welding wire are very similar to those of the parent metal, it is assumed that the mechanical properties of the weld seam and the parent metal are the same.

In order to implement the calculation process of the Ti-6Al-4V solid phase transformation model established in Section 3.3 through ABAQUS (Dassault Systemes, version 6.14, Hangzhou, China), in the process of thermal analysis, the field variables of the welding process are obtained by calling the internal state variables of subroutine USDFLD. The field variables affecting the Ti-6Al-4V phase transition process are mainly the current temperature, cooling/heating rate and historical peak temperature. These three variables are used as the transformation criteria of the phase transition process. The phase transformation equation is called through the program to calculate the volume fraction of the phase transition product. In the mechanical analysis, the field variable of the phase volume fraction defined in the subroutine USDFLD is called through the subroutine UEXPAN. Based on the method in Section 3.2, considering the influence of phase transformations on thermal strain increments and phase transformation volume changes, the product of the thermal expansion coefficient of different phases and the volume fraction of corresponding phases are summed, then the influence of the welding phase transformation on thermal strain and finally on residual stress is considered.

In order to calibrate the parameters of the heat source model, we need to repeatedly compare the finite element calculation results of different heat source parameters with the macro morphology of the welded joint to determine the appropriate parameter values. Figure 10 shows the macro morphology of the welded joints and the calibrated finite element calculation results. The calculated results are in good agreement with the experimental results.

Finally, the calibrated heat source parameters are obtained. The parameters of the double-ellipsoidal heat source model are $a_{f}=4$, $a_{r}=18$, b=2.5, c=3.5, $f_{f}=0.4$ and $f_{r}=1.6$. The parameters of the three-dimensional Gaussian heat source model are H=4, $r_{e}=1.4$ and $r_{i}=1.175$.

## 4. Results and Discussion

### 4.1. Microstructure Observation of Welded Joints

The microstructure of titanium alloy’s parent metal is shown in Figure 11; it is composed of equiaxed α grain and β matrix. The microstructures of different areas (FZ, HAZ and PM) of weldments of TIG welding and LBW are shown in Figure 12 and Figure 13. Because these regions experience different degrees of thermal cycling, the microstructures observed in the FZ, HAZ and PM are different. In the welding process, due to the low thermal conductivity of titanium alloy, the FZ temperature is much higher than the melting point under the effect of the heating cycle, while in the HAZ, the temperature remains below the melting point, and the uneven cooling rate affects the microstructure.

As shown in Figure 12a, at the junction of PM/HAZ and HAZ/FZ of the TIG weldment, it can be found that the β grains in HAZ, FZ and PM are basically equiaxed crystals, but the grain size increases significantly during the transition from PM to FZ. The increase in grain size is usually attributed to the low thermal conductivity of titanium alloys [18] (Karpagaraj et al., 2015). Due to the low thermal conductivity of titanium, the heat retention time is relatively long, and the heat retention at the β phase transition temperature is conducive to β grain growth, so as to coarsen the grains in FZ and HAZ. Gao et al. (2013) also confirmed that the gradient of β grain size of the titanium alloy TIG weldment exists in the weld seam and gradually decreases from the FZ region to the HAZ and PM region.

In Figure 12a, it can also be seen that there are a few acicular structures in the HAZ of the TIG weldment close to FZ, and these acicular structures disappear at the HAZ/PM interface. These may be α′ martensite or acicular α phase. In the zone of HAZ near PM, the lamellar α phase begins to appear, and the acicular α phase becomes coarser. This also confirms that the zone of HAZ near the FZ boundary has a temperature of more than β phase transition point and some retained β phase forms α′ martensite or acicular α phase in the rapidly cooling rate. Squillance et al. (2012) [19] found that the maximum cooling rate occurs at FZ, followed by HAZ. Therefore, compared with the partially transformed HAZ/PM region, it is expected that the proportion of α′ martensite in HAZ region is much larger. According to Elmer et al. (2004) [20], in the HAZ close to the PM matrix, the temperature will not exceed the β phase transition temperature during heating, so this region is quenched below the β transition point temperature, and the equilibrium phase fraction of the β phase in this region is always less than 1. Therefore, due to the partial phase transition, there is no α′ martensite phase in this region and the proportion of retained β phase is high, which is distributed along the transformed α lamella.

Figure 12b shows that there are some needles of martensite α′ and acicular α phase in the FZ of the TIG weldment. This confirms that the non-diffusion phase transition of β→α′ happens at the β grain boundary in FZ. Committee et al. (2004) [21] posited that a slow cooling rate will lead to a high temperature β phase transformation to the α phase; the α phase is flaky, and retained β phase exists in α lamellar generated by phase transformation. Rapid cooling produces a tightly packed needle α phase, and the further increase in the cooling rate leads to fine shapely needle α′ phase formation. The phase is usually considered to be acicular martensite. Elmer et al. (2004) [20] discussed in detail the evolution of β grain and subsequent martensite α′ with time during the rapid cooling of titanium alloy with TIG welding. Elmer stated that during cooling, the β phase remained stable for a short time and then transformed into α′ martensite. He also found that from FZ to HAZ, the grain size of the β phase decreased gradually.

As shown in Figure 13a, the FZ region of the LBW weldment is composed of very fine α′ martensite; almost no β phase grains were found. Compared with the FZ of the TIG weldment, this is due to the lower heat input and relatively higher cooling rate of the FZ of the LBW weldment. For near α type titanium alloy, Zhang et al. (2007) [22] and Sun et al. (2003) [23] found that LBW welded joints consist of α′ martensite phase. Similarly, Gao et al. (2013) [2] found that complete α′ martensite crystal structure was found in the FZ of LBW welding of a Ti-6Al-4V thin plate, and some secondary α colonies were found in the FZ of a TIG weldment growing along α′ martensite. Elmer et al. (2004) [20] explained that if the cooling rate is fast enough to achieve the α′ martensite transition temperature (Ms) immediately, the complete α′ martensite structure would be generated, but no β grains would be occurring. Compared with the FZ of TIG welding, this is probably the reason for the obtaining of a finer complete α′ martensite structure in the FZ of LBW weldments. As shown in Figure 13b, the microstructure of HAZ of LBW welding is also very complex due to the change of microstructure from the FZ/HAZ boundary to the PM/HAZ boundary. The most obvious change is the amount of orthogonal orientation; α′ martensite decreases from the FZ/HAZ boundary to the HAZ/PM boundary. This feature also exists in the HAZ of TIG weldments due to the reduced cooling rate during the transition from the FZ/HAZ to the HAZ/PM interface.

### 4.2. Simulation Results of Phase Volume Fraction Based on Ti6Al4V Phase Transition Model

#### 4.2.1. Simulation Results of Temperature Field

During the finite element calculation, the following variables are defined through the USDFLD subroutine: the current temperature is the field variable NT11 (°C), the cooling/heating rate is defined as the state variable SDV3 (°C/s), the peak temperature is defined as the state variable SDV4 (°C), the field variable FV1 is defined as the β phase volume fraction, the field variable FV2 is defined as the α phase volume fraction and the field variable FV3 is defined as the α′ martensite phase volume fraction. The time evolution of each variable related to the temperature field is calculated by welding thermal simulation. Figure 14 shows the historical peak temperatures of the TIG weldment and LBW weldment. As can be seen in the figure, the temperature exceeding 1650 °C is the fusion zone, and the FZ width of the TIG weldment is significantly wider than that of the LBW weldment; the temperature range of 650~1650 °C is the heat-affected zone, and the heat-affected zone of the TIG weldment is obviously wider than that of the LBW weldment. The simulation results in Figure 14 are in good agreement with the experiment results in Figure 4. The martensite transition temperature of Ti-6Al-4V is 650 °C. The solid-state transformation occurs mainly in the FZ and HAZ, that is, the zone where the peak temperature is higher than 650 °C.

According to the solid-state phase transformation equation of titanium alloy in Section 3.3, the field variables affecting the phase transformation process of Ti-6Al-4V are mainly the current temperature NT11 (°C), cooling/heating rate SDV3 (°C/s) and historical peak temperature SDV4 (°C). These three variables are used as the criteria of the phase transformation to calculate the volume fraction of each phase component in the weld seam during welding. According to the research of Gao et al. (2013) [2], Committee et al. (2004) [21], Elmer et al. (2004) [20] and Squillace et al. (2012) [19], the cooling rate has a great influence on welding phase transformation. Figure 15, Figure 16 and Figure 17 are the distribution cloud diagrams of the current temperature NT11 (°C) and cooling/heating rate SDV3 (°C/s) at different times during welding. Comparing the temperature distribution of NT11 between LBW and TIG welding, it can be seen that the temperature distribution range of LBW is significantly narrower than that of TIG and that the temperature of FZ is much higher than that of TIG welding, which shows that the power density of LBW is much higher than that of TIG.

According to the cooling/heating rate SDV3 of LBW and TIG welding, for both LBW and TIG welding, when the heat source is moving, the cooling zone is behind the heat source and the Accoheating zone is in front of the heat source. The main differences are the range of the heating and cooling areas and the heating and cooling rates. Because the power density of LBW is higher than that of TIG, the heating rate of LBW in front of the heat source is much higher than that of TIG and the heating range of LBW is narrower than TIG. It can be seen from the figure that the maximum cooling rate of LBW is located behind the moving heat source and is greater than 410 °C/s. According to Elmer’s research (2004) [20], when the FZ temperature drops to 980 °C (below the β phase transition point temperature), the non-diffusion phase transition occurs in the FZ, with the β phase forming complete α′ martensite. However, the cooling rate behind the TIG welding heat source is between 20~410 °C/s except for a few areas more than 410 °C/s; partial martensite transformation occurs in the FZ and its nearby HAZ of TIG welding during cooling. The calculation process suitably explains the experiment results in Section 4.1. The amount of α′ martensite in the FZ of LBW is greater than that of TIG welding. It can be seen in Figure 16 that the cooling process of TIG welding is much slower than that of LBW, resulting in coarse grains in the weld seam of TIG welding.

#### 4.2.2. Simulation Results of Phase Volume Fraction

According to the theoretical model of the phase transformation process in Section 3.3, α phase, α′ martensite phase and retained β phase sections would be produced according to different cooling rates for Ti-6Al-4V welding; Figure 18, Figure 19 and Figure 20 show the distribution cloud diagram of the β phase volume fraction (FV1), α phase volume fraction (FV2) and α′ martensitic phase volume fraction (FV3) at different times in the welding. It can be seen from the figure that the change of phase composition only occurs in the FZ and near the HAZ connected with the FZ. The phase change area of the LBW weldment is significantly smaller than that of the TIG welding, which is consistent with the phase change area of the microstructure obtained from the results in Section 4.1.

As can be seen from Figure 18 and Figure 19, in the initial stage of welding, the main process is austenitizing to form the β phase during heating, and then the α phase is formed when cooling. For the LBW, due to the high temperature of the FZ and its surroundings, the temperature does not fall to the martensitic transformation temperature in the initial cooling stage after the heat source moves; β→α phase diffusion transition is what is mainly occurring at this time. For the TIG welding, due to the slow-moving speed of the heat source, the temperature in some areas falls below the martensitic transition point when the heat source still acts on the weldment; the non-diffusion transformation of β→α′ occurs thusly.

According to the β phase volume fraction (FV1) from Figure 20, in the HAZ close to the PM, the high volume fraction of the β phase indicates that a large amount of retained β phase remains in this area during cooling. This is consistent with the research results of Elmer et al. (2004) [20] and the experiment results in Section 4.1. During heating, the temperature will not exceed the β phase transition point temperature; the temperature range is 650~980 °C. During cooling, the cooling rate is not high enough due to a certain distance from the weld center. Therefore, there is no α′ martensitic phase in this area; the proportion of retained β phase is high.

According to the α phase volume fraction (FV2) and α′ martensite phase volume fraction (FV3) from Figure 20. 

The FZ is mainly composed of α′ martensite phase and α phase. The α′ martensite phase volume fraction in the FZ of LBW is greater than that in the FZ of TIG welding, which is consistent with the microstructure in Section 4.1, which confirms that non-diffusion transformation occurs during rapid cooling to form the α′ martensite phase, and diffusion transformation occurs during slow cooling to form the α phase.

Figure 21 shows the TIG welding numerical simulation results of the change of the volume fractions of the α phase, β phase and α′ phase under the welding thermal cycle in the weld seam 50 mm away from the starting position. Figure 21a shows the phase change of the FZ center, and Figure 21b shows the phase change of the PM/HAZ interface. The initial microstructure consists of about 0.9247 α and 0.07526 β phase. During the heating, with the rapid increase in temperature, the α phase fraction decreased rapidly to 0, while the β phase increased to 1. In fact, the β phase should also become 0 above the melting temperature, but for simplicity, it is still maintained at 1 above the β transition point temperature. When it is cooled below the β phase transition point temperature, the β phase forms different product phases through diffusion phase transition or non-diffusion phase transition according to the cooling rate. As can be seen from Figure 21a, during cooling, the β phase is first transformed into the α phase, and after cooling to the martensitic transformation temperature, the β phase begins to transform into the α′ martensitic phase; with the continuous decrease in the cooling rate, the β phase is transformed into the α phase again until the volume fraction reaches the equilibrium phase of the β phase, then the composition of each phase remains unchanged. With cooling to the room temperature, the final phase is mainly composed of 0.479α, 0.482α′ and 0.039β phases. As can be seen from Figure 21b, the PM/HAZ zone is far from the center of the FZ, the temperature is not high during heating and the cooling rate is slow, resulting in a large amount of retained β phase; the final phase is composed of 0.309α, 0.0α′ and 0.691β phases.

Figure 22 shows the LBW numerical simulation results of the change of the volume fraction of the α phase, β phase and α′ phase in the weld seam 50 mm away from the starting position. Figure 22a shows the phase change of the FZ center, and Figure 22b shows the phase change of the PM/HAZ interface. The β phase first transformed into the α phase. After cooling to a certain temperature, the β phase transformed into the α′ martensite phase. The final phase is mainly composed of 0.569α, 0.385α′ and 0.046β phases. As can be seen from Figure 22b, due to the concentrated power density of the LBW heat source, the temperature in the PM/HAZ zone decreases rapidly and the cooling rate is relatively slow, resulting in a certain amount of retained β phase.

### 4.3. Verification of Consistency of Hardness Distribution and Phase Volume Fraction Distribution of Welded Joints 

The significant fluctuation of Vickers hardness is mainly due to the differences in grain size and phase composition observed in FZ, HAZ and PM, as well as the tight packing structure of titanium obtained by Liu et al. (2012) [24]. According to Pasang et al. (2013) [25], the hardness-increasing mechanism of the FZ of the α and α + β titanium alloy is almost the same as the formation of the α′ phase substructure. Gao et al. (2013) [2], Squillace et al. (2012) [19] and X. Ã. Cao et al. (2009) [26] found that the increase in hardness was due to the appearance of the α′ martensitic phase and acicular phase in the FZ and HAZ of titanium alloy weldments. According to Zeng and Bieler (2005) [27], the hardness of these phases follows the order: α′ martensite > acicular α > β.

Table 4 and Table 5, respectively, show the measured values of micro-hardness distribution of TIG weldment and LBW weldment from weld centerline to PM; the measuring instrument was the VMH-I04 micro-hardness tester. In order to analyze the relationship between volume fraction of phase composition and hardness distribution, the calculated results of phase volume fraction in Section 4.2.2 and the measured values of micro-hardness distribution are compared with plotting. The hardness change and volume fraction of phase composition from weld centerline to PM in the TIG and LBW weldments are shown in Figure 23.

The Vickers hardness relationship of the weld area is FZ > HAZ > PM. The hardness value of the FZ of the LBW weldment is about 54.57 HV higher than that of the PM. There is a narrow HAZ area of about 0.75 mm between the FZ and PM of the LBW weldment, which has a very large hardness gradient of 72.76 HV/mm. For TIG weldments, the HAZ area is wider, and the gradient of hardness reduction from the FZ to PM is about 3.66 HV/mm, which is much smaller than that of the LBW-welded joints. 

It can be found from the figure that the maximum Vickers hardness appears in the FZ, the FZ hardness of the LBW weldment is significantly higher than that of the TIG weldment and the volume fraction of the α′ martensite phase in the FZ of the LBW weldment is higher than that of the TIG weldment. According to the order of phase hardness, the hardness mainly depends on the α′ martensite phase, and the formation of α′ martensite phase depends on the cooling rate. By comparing Figure 12b and Figure 13a, it can be found that the acicular α′ phase of the FZ of the LBW weldment is finer than that in the FZ of the TIG weldment. Therefore, it can be concluded from the experimental and simulation results that a higher cooling rate leads to more α′ phase formation and a higher hardness value.

It can be seen from the figure that the α′ martensite phase in the HAZ region of the TIG weldment and the LBW weldment gradually decreases; mainly composed of the α phase and β phase, the phase composition distribution is uneven. However, the hardness value of the HAZ area of the TIG weldment is slightly higher than that of the LBW weldment, and the grain size of the TIG weldment is also larger than that of the LBW weldment, indicating that the grain size has a certain influence on the hardness value distribution. 

Through the above analysis, it can be considered that the hardness change is mainly due to the formation of acicular α phase, which is composed of α′ martensite and acicular α phase formed with rapid cooling, while the effect of grain size on hardness increasing from the PM to the FZ is not particularly obvious.

### 4.4. Verification of Simulation Results of Residual Stress and Deformation Considering Solid-State Phase Transformation

#### 4.4.1. Influence of Welding Technology on Residual Stress

Commin et al. (2012) [28] state that residual stresses in welded structures may lead to early yield, so they have a significant impact on the mechanical properties of components. When the heat source moves on the titanium alloy plate, the weld seam will undergo rapid heating to cooling, which will lead to uneven expansion and contraction of the weld seam metal. At the same time, rapid cooling makes the microstructure of the weld seam change significantly. Liu et al. (2012) [24] state that grain coarsening, substructure strengthening and solid solution strengthening in the FZ and HAZ are caused by rapid cooling. During welding, the thermal and complex microstructure evolution makes the distribution of residual stress more complex in the weld seam.

The research on the residual stress of the LBW weld seam by Zhang et al. (2005) [3] and Liu et al. (2009) [29] shows that the stress gradient near the weld center is very steep and that the tensile stress near the weld center changes to compressive stress within the first few millimeters. Compared with the stress in TIG weldment, the stress distribution in LBW weldment near the weld center has a steeper gradient. This is related to the higher energy density of LBW welding than TIG welding. It can be seen from Figure 24 that the peak stress in the weld center of the LBW weldment is between 900~1000 MPa, while that of the TIG weldment is not more than 750 MPa, which is mainly due to the fact that the plastic deformation of the LBW welding in the weld seam is greater than that of the TIG welding. Along the depth direction, the residual stress distribution of TIG welding has a large gap. The surface residual stress is about 150 MPa greater than that at the depth of 0.8 mm, while the difference in the LBW is small. This is because the heat source energy of LBW welding is more concentrated and evenly distributed than that of TIG welding, so the difference of residual stress along the depth direction is small.

Figure 25 compares the longitudinal stress distribution on the surface of the LBW weldment and TIG weldment and at a depth of 0.8 mm. Thermal strain has an important influence on the distribution of welding residual stress, but the phase transformation during cooling also affects the magnitude of residual stress, especially in the zone near the weld center. Appolaire et al. (2015) [30] reported that during cooling, formation of α′ martensite or transformation from the β phase to α phase leads to an increase in the local volume of the material. Comparing the zone where the longitudinal residual stress changes from tensile stress to compressive stress, the compressive stress of TIG weldment is −150~−250 MPa and LBW is −50~−100 MPa. TIG welding produces greater compressive stress in the HAZ. This is because there are more retained β phases in the HAZ, resulting in the volume shrinkage of the zone and the volume expansion of the FZ due to the generation of a large amount of α′ martensite phase. These compressive stresses reduce the effect of tensile stress caused by thermal strain near the weld center.

The transverse stress distribution of TIG and LBW weldments is shown in Figure 26. It can be observed that for the TIG and LBW weldments, the overall magnitude of transverse stress is significantly smaller than that of longitudinal stress. The main reason for this is that there is a very sharp temperature gradient along the weld direction (longitudinal). The energy density of LBW welding is higher than that of TIG welding, and its plastic zone is narrow and concentrated, while the plastic deformation zone of TIG is much larger. In addition, the welding heat input of TIG welding is larger, resulting in larger deformation of TIG weldment, so the plastic deformation of TIG welding in the weld seam is released to a certain extent; thus, its peak stress is less than that of LBW weldment. The variation of residual stress along the thickness direction is mainly caused by plastic strain and deformation. The residual stress fluctuation in the HAZ is mainly caused by a large amount retained β phase in the HAZ, resulting in the volume shrinkage.

Solid phase transformation mainly occurs in the FZ and HAZ zone, which changes the phase composition and phase proportion of the weld seam. Due to the different thermal expansion coefficients and different unit lattice volumes of each phase, solid phase transformation affects the local expansion and contraction of the weld seam and then changes the distribution of tensile and compressive stress in the local of the weld seam, but the solid phase transformation has little effect on the overall distribution trend of residual stress.

#### 4.4.2. Influence of Welding Technology on Deformation

Murakawa (2013) [31] reported that welding deformation and residual stress are strongly affected by welding heat input. Short (2009) [1] reported that LBW welding has higher power density than TIG welding. Therefore, in order to achieve penetration in materials of the same thickness, TIG welding provides more heat input and produces greater stress and deformation. Figure 27 shows the overall deformation distribution of TIG and LBW weldments. Figure 28a shows the bending deformation along the longitudinal centerline (along the welding direction) of the two weldments, and Figure 28b shows the bending deformation along the transverse centerline (perpendicular to the welding direction) of the two weldments.

As shown in Figure 27, the TIG weldment has a large bending deformation in longitudinal and transverse directions, while the LBW weldment has a small deformation, mainly because the total welding heat input of the TIG weldment is greater than that of the LBW weldment, and the heating zone of LBW is extremely narrow. As shown in Figure 28a, the maximum bending deformation of TIG welding is about 2 mm, and that of LBW welding is about 0.25 mm. The transverse deformation observed in Figure 28b is concave in the center and warped on both sides to form a concave bend. In the FZ, both the TIG weldment and LBW weldment have intermediate bulge deformation. The solid phase transformation mainly occurs in the weld seam. The phase transformation changes the phase composition and phase proportion in the weld seam. The unit lattice volume of each phase is different, resulting in local volume changes. It can be seen from Figure 27 and Figure 28 that the local volume changes caused by phase transformation have little effect on the overall deformation, which is mainly determined by the heat input of different welding processes.

## 5. Conclusions

By summarizing the current research results of TIG and LBW welding processes and solid-state phase transformation models of Ti-6Al-4V, a mathematical model of Ti-6Al-4V solid-state phase transformation suitable for welding was established by using the JMAK equation and K–M equation, and a thermal-phase-mechanics coupling finite element calculation method considering solid-state phase transformation was developed by using ABAQUS software to realize the numerical simulation of different welding processes (LBW and TIG welding). Through the numerical simulation calculations and experimental verification of different welding processes (LBW and TIG welding) for Ti-6Al-4V titanium alloy sheets, the following conclusions were obtained:(1)Although the distribution of phase composition and phase volume fraction of LBW- and TIG-welded joints by numerical simulation cannot fully reflect the complexity of microstructure evolution, it can express the approximate distribution trend of phases: For the LBW welded joint, the FZ and FZ/HAZ interfaces are mainly martensite α′ phase, while the HAZ/PM interface retains a certain amount of residue β phase. For the TIG-welded joint, the FZ is partial martensite α′ phase, and the HAZ/PM interface retains a large amount of residual β phase. Therefore, the numerical method established in this paper can obtain useful information of phase composition and distribution of welded joints with different welding processes (LBW and TIG welding) in the form of scalar fields.(2)Because the thermal expansion coefficient and unit cell volume of the α phase and β phase are different, the numerical simulation results of residual stress and deformation will be affected after considering the phase composition and phase volume fraction distribution of welded joints. For LBW- and TIG-welded joints, the phase transformation process from the β phase to the martensitic α′ phase or to the α phase is a process of material volume expansion. Therefore, during welding, the FZ area expands and the HAZ/PM area shrinks, resulting in tensile stress in the FZ area and compressive stress in the HAZ/PM narrow area.(3)In extension, through the research results of LBW- and TIG-welded joints, the scalar fields of the phase volume fraction and residual stress can be used as the characteristic quantities of the welding process, and the scalar fields of the phase volume fraction and residual stress can be introduced into the numerical analysis of structural fracture failure as welding process factors, so the influence of the welding process on structural fracture failures can be considered.

## Figures and Tables

**Figure 1 materials-15-02882-f001:**
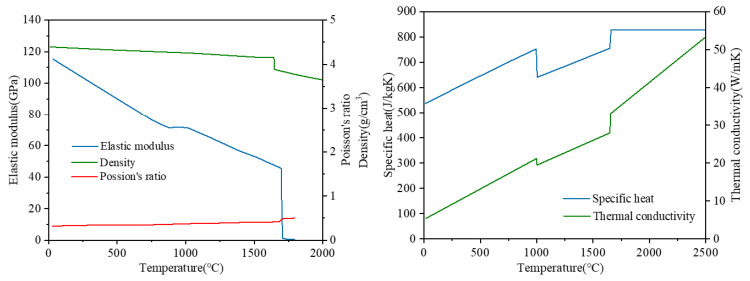
Thermo-physical properties of Ti-6Al-4V with temperature.

**Figure 2 materials-15-02882-f002:**
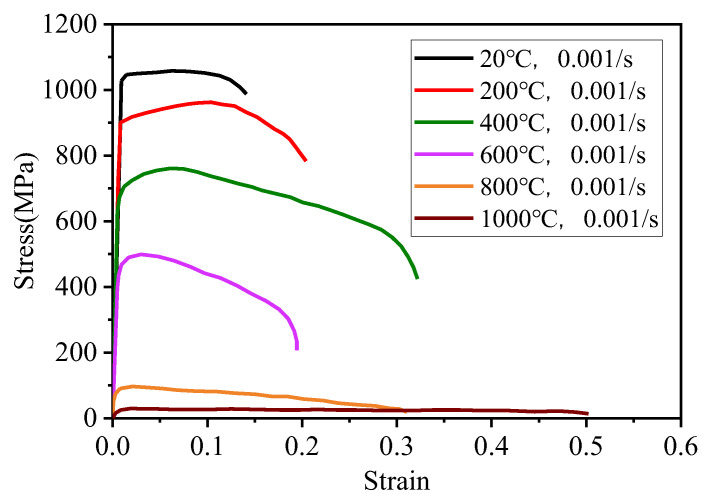
Stress–strain curve of Ti-6Al-4V with temperature [9].

**Figure 3 materials-15-02882-f003:**
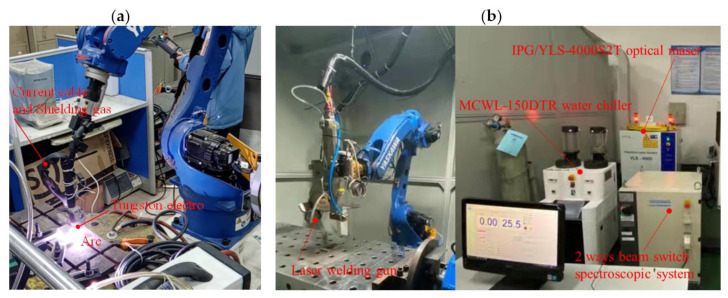
Welding process of Ti6Al4V titanium alloy: (**a**) TIG welding, (**b**) LBW.

**Figure 4 materials-15-02882-f004:**
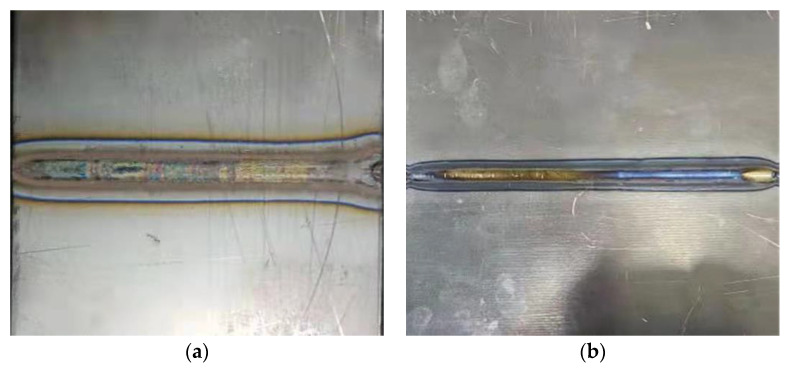
Upper surface appearance of titanium alloy weldment: (**a**) TIG weldment, (**b**) LBW weldment.

**Figure 5 materials-15-02882-f005:**
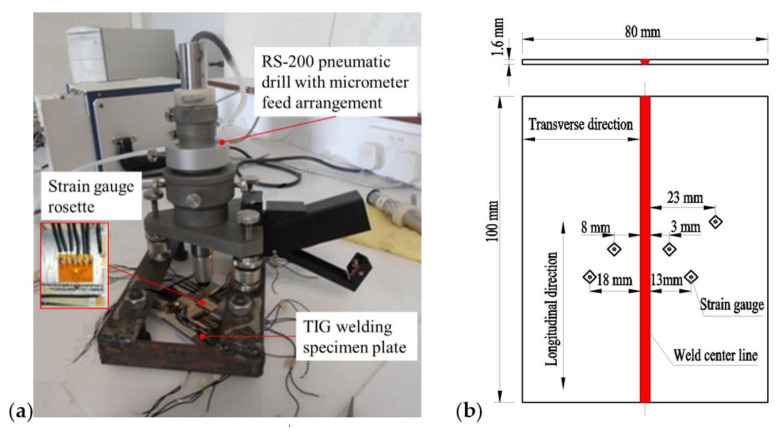
Measurement of residual stress by drilling method. (**a**) Measuring device, (**b**) measuring residual stress location.

**Figure 6 materials-15-02882-f006:**
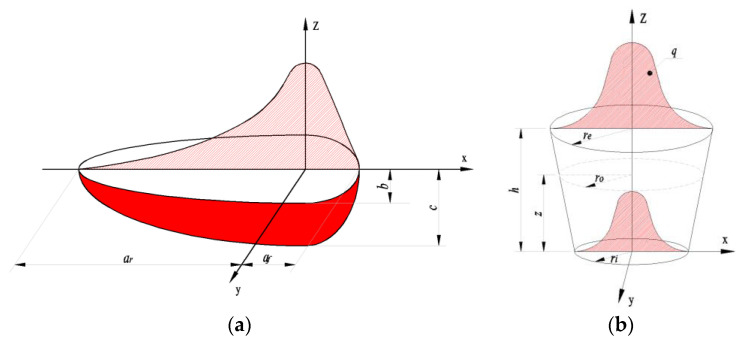
Heat source model of welding process (unit: mm). (**a**) Double-ellipsoidal heat source model. (**b**) Cone heat source model.

**Figure 7 materials-15-02882-f007:**
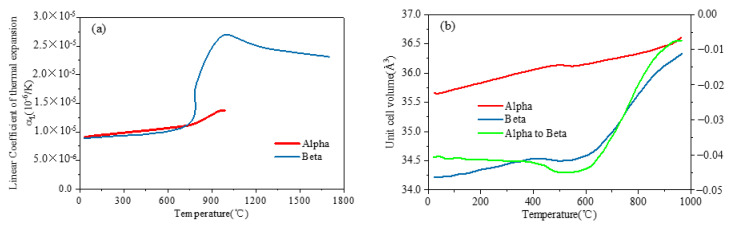
(**a**) Linear coefficient of thermal expansion; (**b**) unit cell volume for each phase, and transformation-induced volumetric strains as a function of temperature [5].

**Figure 8 materials-15-02882-f008:**
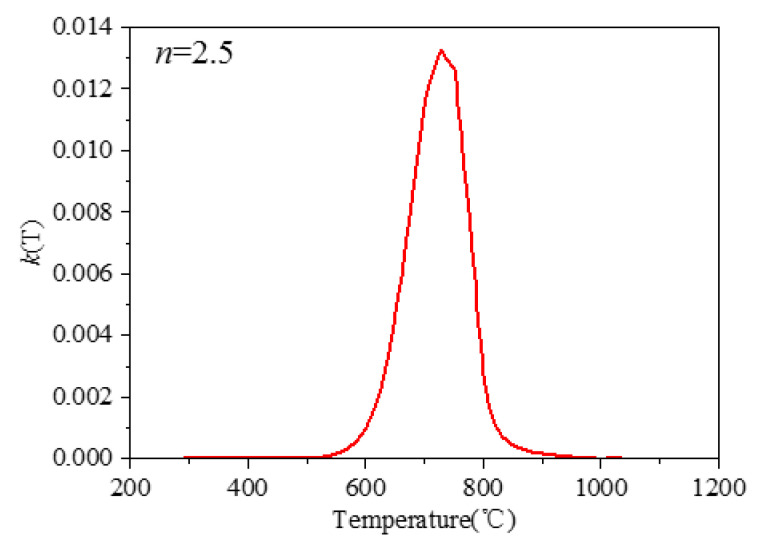
Kinetic parameters $k_{\beta \to \alpha }$ of Ti-6Al-4V diffusion-controlled transition.

**Figure 9 materials-15-02882-f009:**
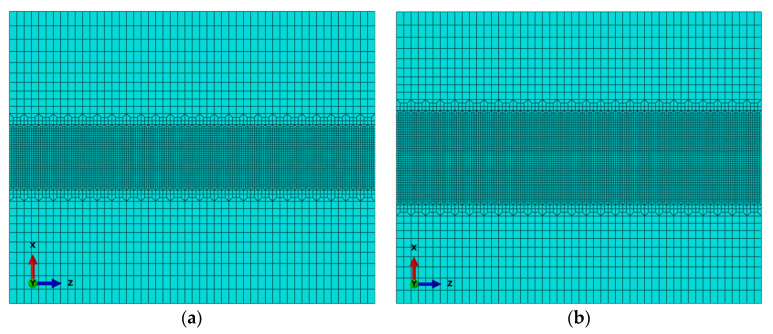
Welding finite element model: (**a**) LBW, (**b**) TIG welding.

**Figure 10 materials-15-02882-f010:**
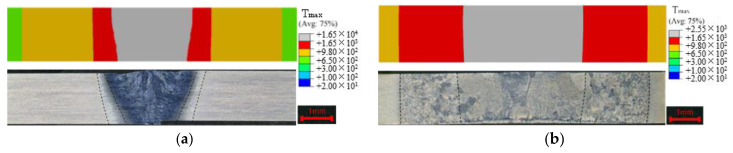
Comparison between the experimental and the simulated weld profiles: (**a**) LBW, (**b**) TIG welding.

**Figure 11 materials-15-02882-f011:**
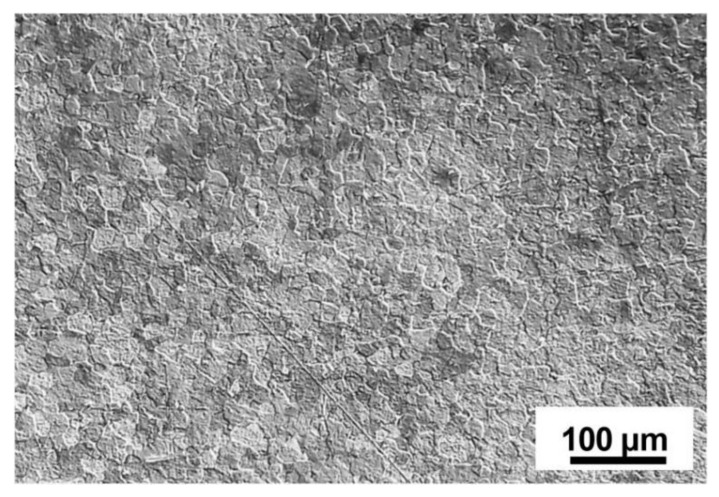
Microstructure of base alloy showing equiaxed α grains in prior β matrix.

**Figure 12 materials-15-02882-f012:**
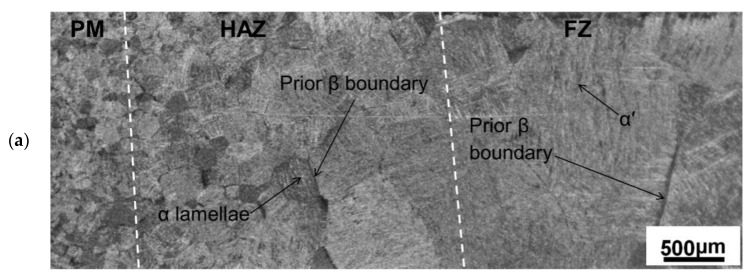
Microstructures of different zones of TIG weldment: (**a**) PM/HAZ/FZ interface and (**b**) FZ.

**Figure 13 materials-15-02882-f013:**
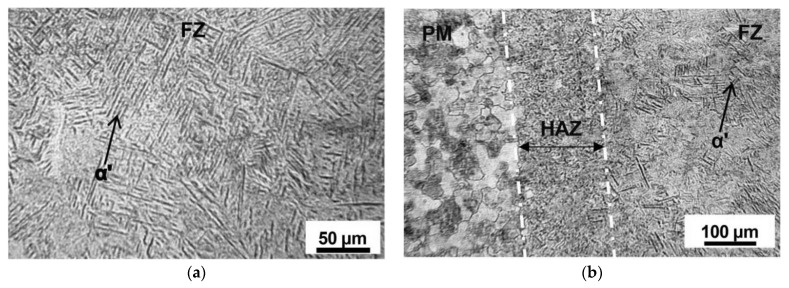
Microstructures of different zones of LBW weldment: (**a**) FZ, (**b**) FZ/HAZ/FM interface.

**Figure 14 materials-15-02882-f014:**
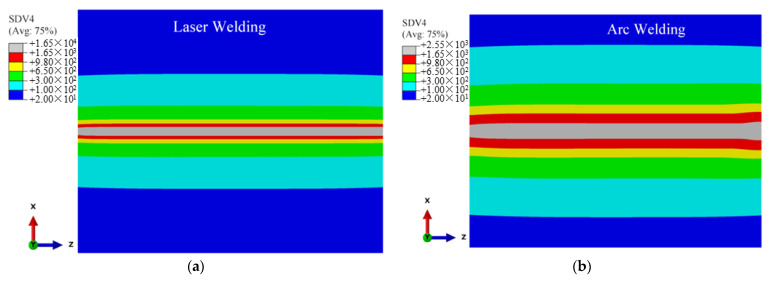
Historical peak temperature of welding process: (**a**) LBW weldment, (**b**) TIG weldment.

**Figure 15 materials-15-02882-f015:**
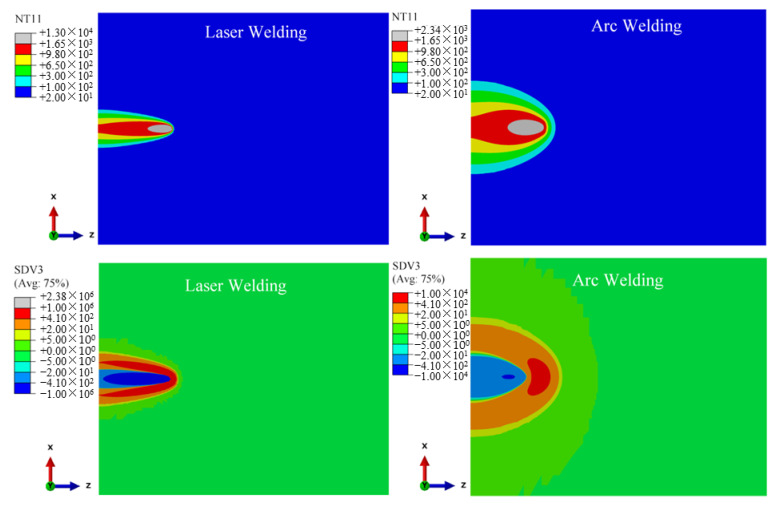
25 mm from starting position, changes of current temperature NT11(°C) and cooling/heating rate SDV3(°C/s) during welding.

**Figure 16 materials-15-02882-f016:**
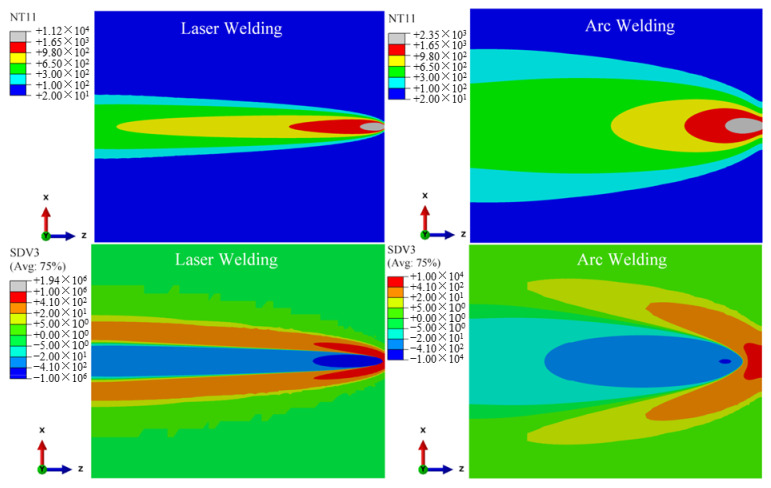
100 mm from starting position, changes of current temperature NT11(°C) and cooling/heating rate SDV3(°C/s) during welding.

**Figure 17 materials-15-02882-f017:**
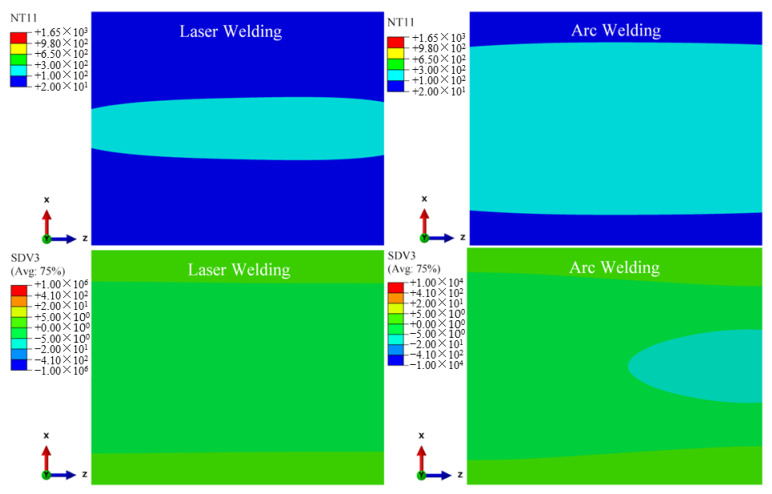
60 s from start time, changes of current temperature NT11(°C) and cooling/heating rate SDV3(°C/s) during welding.

**Figure 18 materials-15-02882-f018:**
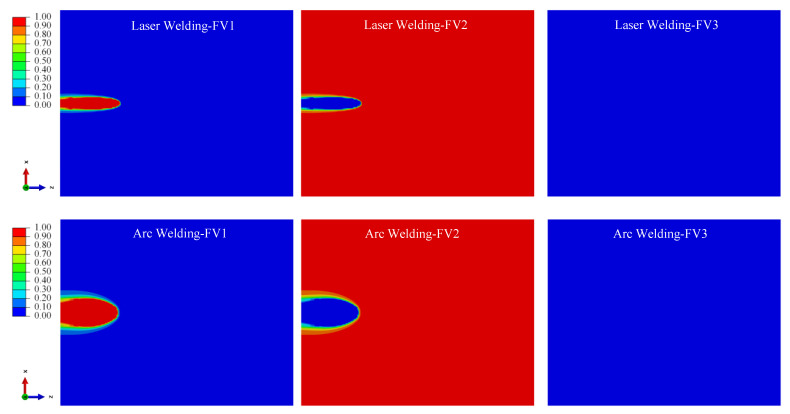
With the heat source moving to 25 mm from starting position, change of volume fraction of phase composition in weld bead.

**Figure 19 materials-15-02882-f019:**
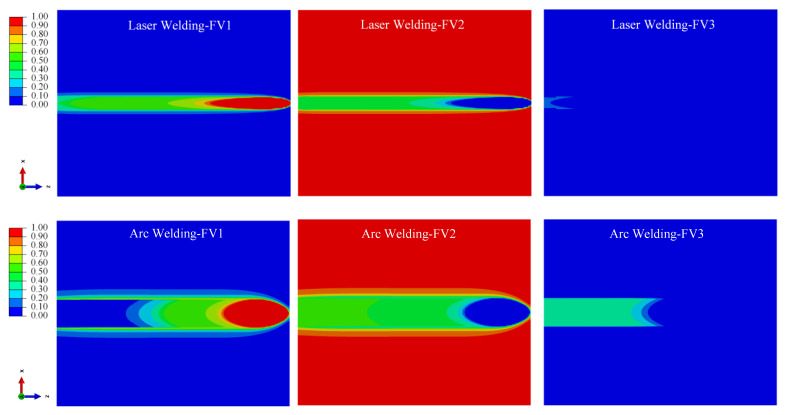
With the heat source moving to 100 mm from starting position, change of volume fraction of phase composition in weld bead.

**Figure 20 materials-15-02882-f020:**
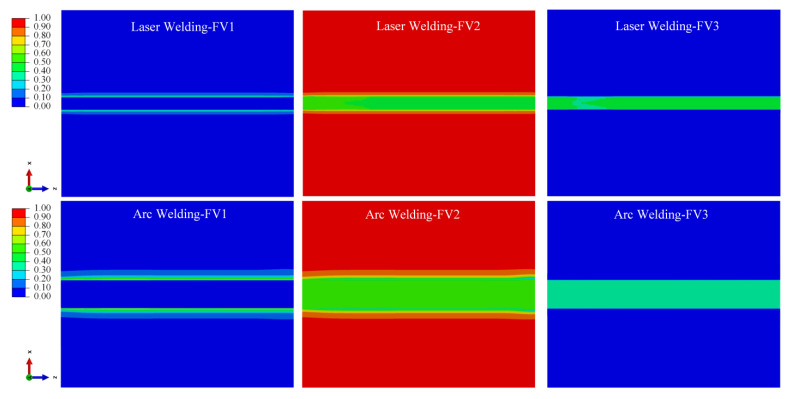
With 60 s from the start time, change of volume fraction of phase composition in weld bead.

**Figure 21 materials-15-02882-f021:**
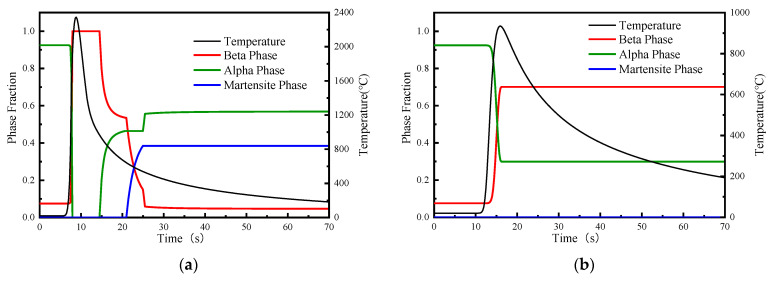
Change of phase volume fraction in weld area of TIG weldment: (**a**) FZ zone, (**b**) PM/HAZ interface.

**Figure 22 materials-15-02882-f022:**
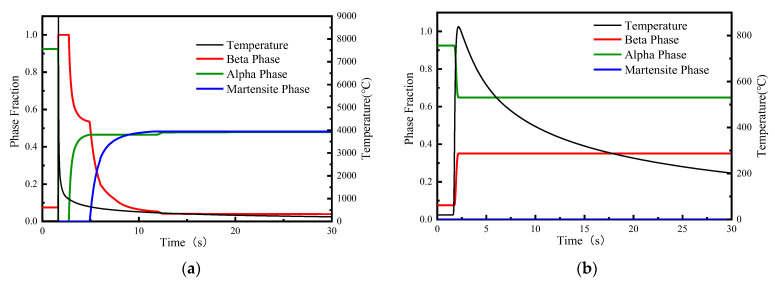
Change of phase volume fraction in weld area of LBW weldment: (**a**) FZ zone, (**b**) PM/HAZ interface.

**Figure 23 materials-15-02882-f023:**
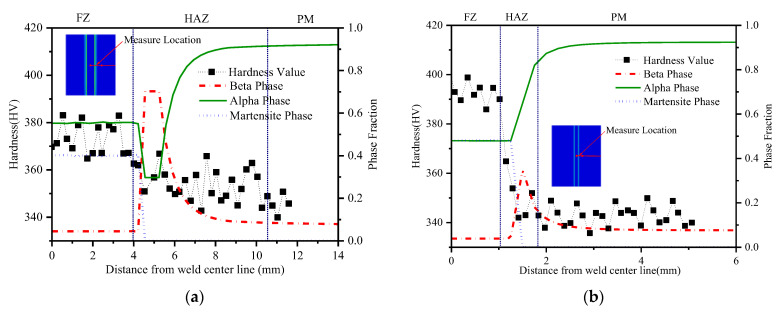
Micro-hardness distribution: (**a**) TIG weldment, (**b**) LBW weldment.

**Figure 24 materials-15-02882-f024:**
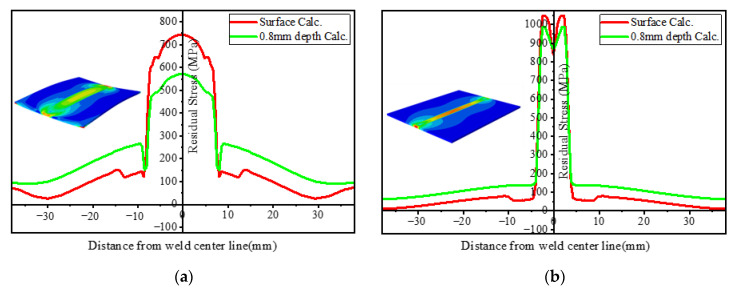
Mises stress distribution of residual stress at different depths: (**a**) TIG weldment, (**b**) LBW weldment.

**Figure 25 materials-15-02882-f025:**
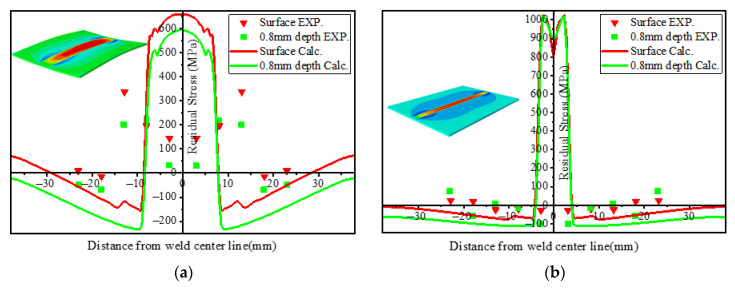
Longitudinal residual stress distribution at different depths: (**a**) TIG weldment, (**b**) LBW weldment.

**Figure 26 materials-15-02882-f026:**
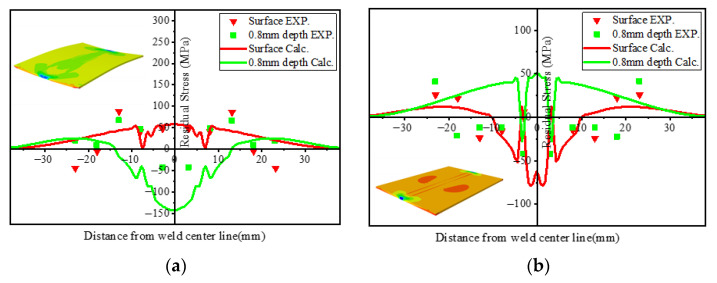
Transverse residual stress distribution at different depths: (**a**) TIG weldment, (**b**) LBW weldment.

**Figure 27 materials-15-02882-f027:**
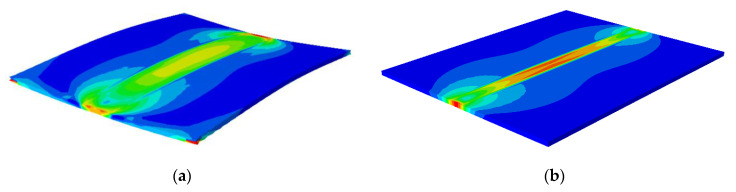
Experimental and simulation results of butt welding: (**a**) TIG weldment, (**b**) LBW weldment.

**Figure 28 materials-15-02882-f028:**
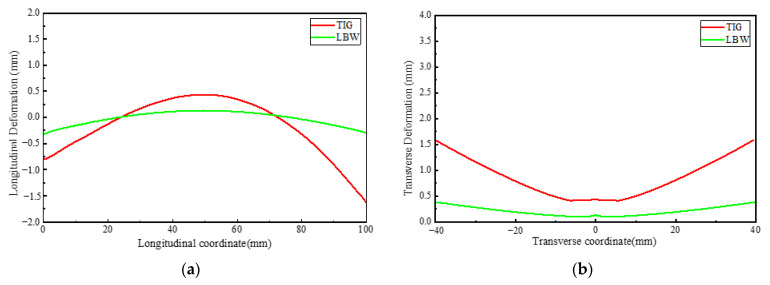
(**a**) Longitudinal deformation, (**b**) transverse deformation.

**Table 1 materials-15-02882-t001:** Chemical composition of Ti-6Al-4V titanium alloy (mass fraction%).

Ti	Al	V	Fe	C	N	H	O
rest	6.5	4.4	0.30	0.10	0.05	0.015	0.20

**Table 2 materials-15-02882-t002:** Welding parameters for TIG welding and the dimensions of the weld seam.

Current		Pulse Width		Voltage (V)	Welding Speed (mm/min)	Arc Length (mm)	Shielding Gas	Top Width of FZ (mm)	Bottom Width of FZ (mm)
Primary (A)	Background (A)	High (ms)	Low (ms)						
32	16	8	4	10	32.5	4	Argon	5.2	3.8

Note: “Top Width of FZ” delineates the top width of the fusion zone of the weld seam section. “Bottom Width of FZ” delineates the bottom width of the fusion zone of the weld seam section.

**Table 3 materials-15-02882-t003:** Welding parameters for LBW and the dimensions of the weld seam.

Power (kW)	Pulse Duration (ms)	Pulse Frequency (Hz)	Welding Speed (mm/min)	Defocus Distance (mm)	Shielding Gas	Top Width of FZ (mm)	Bottom Width of FZ (mm)
0.8	8	8	160	0	Argon	2.35	1.75

Note: “Top Width of FZ” delineates the top width of the fusion zone of the weld seam section. “Bottom Width of FZ” delineates the bottom width of the fusion zone of the weld seam section.

**Table 4 materials-15-02882-t004:** Measurement values of micro-hardness distribution of TIG weldment starting from weld center.

Distance (mm)	0	0.5	1.0	1.5	2.0	2.5	3.0	3.5	4.0	4.5	5.0	5.5
Hardness (HV)	369.8	383.1	369.2	382.1	367.0	367.1	377.2	366.9	362.7	350.9	356.8	358.0
Distance (mm)	6.0	6.5	7.0	7.5	8.0	8.5	9.0	9.5	10.0	10.5	11.0	11.5
Hardness (HV)	349.8	355.7	357.8	365.8	359.0	349.0	345.0	360.2	357.1	348.8	339.9	345.7

**Table 5 materials-15-02882-t005:** Measurement values of micro-hardness distribution of LBW weldment starting from weld center.

Distance (mm)	0	0.25	0.5	0.75	1.0	1.25	1.5	1.75	2.0	2.25	2.5
Hardness (HV)	392.9	398.8	394.7	394.6	364.8	342.0	351.9	337.9	344.0	339.8	342.9
Distance (mm)	2.75	3.0	3.25	3.5	3.75	4.0	4.35	4.5	4.75	5.0	
Hardness (HV)	343.9	337.6	343.9	343.9	343.9	349.9	340.1	348.7	338.6	339.9	

## Data Availability

Not applicable.

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
