# Peer review of "A Numerical Simulation Method Considering Solid Phase Transformation and the Experimental Verification of Ti6Al4V Titanium Alloy Sheet Welding Processes"

_materials, 2022, doi:10.3390/ma15082882_

Round 1
Reviewer 1 Report
The article in the present form is quite good in content and quality. The style of reference and reference list need to be corrected following the journal recommendation.
Author Response
Dear Professor, I'm very sorry for the long delay in replying to your review comments, because it is necessary to supplement the macro section of the weld joints, but the equipment in the laboratory happened to be out of order. So I hope to get your understanding.
I have corrected the citation format, please refer to the updated version manuscripts.
Reviewer 2 Report
Dear Authors,
I have read the results of the study with great attention: "A Comparative Study on Numerical Simulation and Experiment of Laser and TIG Welding of Titanium Alloy Sheet". In terms of content, the presented study is correct, but in order for it to be published, some corrections and additions are necessary.
Detailed comments:
- In the literature analysis, it is worth using studies on silumation using the Sysweld program (e.g. works by T. Kik)
- How was the chemical composition of the tested alloy determined, especially the content of C, N, O, H
- Please present the welding parameters using the TIG method and the laser beam using the tables
- How sheets were prepared for welding?
- What shielding gas was used?
- Was a shielding drag nozzle used in TIG welding?
- Throughout the article, the values given in the formulas are shown in the upper index - I do not see any justification for this spelling
- There is no simulation result for TIG and laser so as to show that it has been calibrated and not only information that the correct values were obtained as a result of a trial analysis.
- Please supplement the work with macroscopic photos of welded joints
- Please enter the load at which the hardness was measured
Author Response
Dear Professor, I'm very sorry for the long delay in replying to your review comments, because it is necessary to supplement the macro section of the weld joints, but the equipment in the laboratory happened to be out of order. So I hope to get your understanding.
I have completed your reply, please refer to the updated version of the attachment (author-coverletter-16966071.v1.doc) and manuscripts for detailed reply.

Reviewer 3 Report
Extensive experiments and numerical simulation were carried out to study butt welding of thin Ti6Al4V plates. The paper has numerous problems and only the major problems are highlighted here:
(1) The title is misleading and must be improved.
The major objective is to compare TIG welding and laser welding. It is not "a comparative study on numerical simulation and experiment". Numerical simulation and experiment are merely means for the comparative study of TIG welding and laser welding. Only one titanium alloy was studied, so this is not a general study of titanium alloy. The name of the alloy should be specified in the title.
(2) The paper is unnecessarily long. It is so long because it contains too much description of basic knowledge. For example, the description on laser welding in Section 2.2 is entirely useless for a research paper. Strangely, useful information on the laser welding experiments is completely missing. Which type of laser was used? What was the laser wavelength? Was it a continuous wave laser or pulsed laser. Without such essential information the study becomes meaningless. Similar problem exists in the part of description on TIG welding. None of the essential information on welding current, voltage and velocity is given. The description on TIG welding in line 167 is completely wrong.
(3) Many names of the authors in the references are missing, as can be seen from References 10, 11 and 12. This is perhaps not a major problem, but it is very disturbing.
(4) A good research paper must possess value proposition. It should give readers new knowledge rather than tell them what they have already known. Unfortunately, your conclusion number 3 contains statements well known to welding researchers.
Author Response
Dear Professor, I'm very sorry for the long delay in replying to your review comments, because it is necessary to supplement the macro section of the weld joints, but the equipment in the laboratory happened to be out of order. So I hope to get your understanding.
I have completed your reply, please refer to the updated version of the attachment (author-coverletter-16966094.v1.doc) and manuscripts for detailed reply.

Round 2
Reviewer 2 Report
Dear Authors,
Thank you for addressing all my comments and suggestions. I believe that the article can be published as it is.
Author Response
Dear Professor, thank you for your valuable comments and academic guidance. According to the comments of the academic editor, the manuscript has been modified and replied. Please refer to the coverletter and manuscript for detailed modifications. I wish you a happy life, a happy mood every day.
